# Urinary Comprehensive Genomic Profiling Correlates Urothelial Carcinoma Mutations with Clinical Risk and Efficacy of Intervention

**DOI:** 10.3390/jcm11195827

**Published:** 2022-09-30

**Authors:** Vincent T. Bicocca, Kevin G. Phillips, Daniel S. Fischer, Vincent M. Caruso, Mahdi Goudarzi, Monica Garcia-Ransom, Peter S. Lentz, Andrew R. MacBride, Brad W. Jensen, Brian C. Mazzarella, Theresa Koppie, James E. Korkola, Joe W. Gray, Trevor G. Levin

**Affiliations:** 1Convergent Genomics, Inc., South San Francisco, CA 94080, USA; 2Willamette Urology, Salem, OR 97302, USA; 3Department of Biomedical Engineering, Knight Cancer Institute, MEP-LINCS Center, Oregon Health & Science University, Portland, OR 97239, USA

**Keywords:** bladder cancer, next generation sequencing, machine learning, genomic profiling, liquid biopsy, urine

## Abstract

The clinical standard of care for urothelial carcinoma (UC) relies on invasive procedures with suboptimal performance. To enhance UC treatment, we developed a urinary comprehensive genomic profiling (uCGP) test, UroAmplitude, that measures mutations from tumor DNA present in urine. In this study, we performed a blinded, prospective validation of technical sensitivity and positive predictive value (PPV) using reference standards, and found at 1% allele frequency, mutation detection performs at 97.4% sensitivity and 80.4% PPV. We then prospectively compared the mutation profiles of urine-extracted DNA to those of matched tumor tissue to validate clinical performance. Here, we found tumor single-nucleotide variants were observed in the urine with a median concordance of 91.7% and uCGP revealed distinct patterns of genomic lesions enriched in low- and high-grade disease. Finally, we retrospectively explored longitudinal case studies to quantify residual disease following bladder-sparing treatments, and found uCGP detected residual disease in patients receiving bladder-sparing treatment and predicted recurrence and disease progression. These findings demonstrate the potential of the UroAmplitude platform to reliably identify and track mutations associated with UC at each stage of disease: diagnosis, treatment, and surveillance. Multiple case studies demonstrate utility for patient risk classification to guide both surgical and therapeutic interventions.

## 1. Introduction

Urothelial carcinoma (UC) of the bladder is the sixth most common cancer in the United States, with an estimated 83,730 new cases in 2021 [1]. Approximately 75% of new cases are diagnosed in individuals 65 years of age and older [1], and diagnosis is typically made following bouts of blood in the urine (hematuria) or painful urination. Following treatment, patients with both low- and high-risk disease undergo intensive screening regimens of transurethral cystoscopies to visually inspect the bladder for disease recurrence every 3–6 months for at least 5 years and annually thereafter [2,3]. Aggressive post-treatment screening is required as the current standard of care results in a 60–70% recurrence rate following initial treatment, and with disease progression in 5–20% of patients [4]. Despite high recurrence rates, screening by cystoscopy has low compliance [5]. The ongoing challenges of implementing effective risk-stratified patient care results in UC having one of the highest average life-long per-patient surveillance costs of any solid cancer [6,7].

A urine-based diagnostic that detects tumors with high performance and provides prognostic risk insights would enhance patient standard of care. Currently, urine cytology serves as the standard noninvasive test, but performs poorly in the detection of low-grade disease, and has poor overall sensitivity and a high “atypical/indeterminate” (uninformative) test result rate [8,9,10,11]. Multiple noninvasive urine biomarker tests have come to market with similar limitations [12,13,14,15]. These tests are characterized by sparse measurements of cellular processes (like proliferation) that can act as a proxy for tumorigenesis but are also fundamental biological processes, associated with tissue damage/repair (e.g., kidney and bladder stones, chronic kidney disease, catheterization, and recent instrumentation) and inflammation (bacteriuria, cystitis, chronic urinary retention). These issues make biomarker testing highly susceptible to false-positive test results in a clinical setting where these confounding factors are prevalent. As a result, these technologies offer improved sensitivity over cytology at the cost of compromised specificity [16,17]. In general, biomarker urine tests have failed to achieve widespread clinical adoption due to a failure to provide additional prognostic insights, middling performance [18], and a lack of quality clinical benefit studies [19].

Advances in next-generation sequencing (NGS) technologies have facilitated the growth of nucleic acid-based diagnostic strategies to quantify disease through expansive genomic measurements. Assays that characterize circulating tumor DNA in blood plasma have moved the concept of “liquid biopsy” into the clinic [20]. FDA-approved NGS technologies now support clinical decision making for multiple cancer types, informing disease diagnosis [21], prognosis [22], treatment selection [23,24,25], recurrence monitoring [26,27], and prediction of therapy resistance mechanisms [28,29]. Beyond blood, nascent urine-based liquid biopsy technologies have been developed to analyze tumor and cell-free DNA for somatic mutations [30,31,32] and DNA methylation signatures [33,34,35]. These research-grade technologies offer promise for noninvasive detection of urologic tract malignancies; however, crucial obstacles remain to achieve a robust clinical test. Fundamental challenges persist around the variability of urine, the costs and timing of NGS and laboratory processing, the complexity and sophistication of algorithm customization, and the large studies required for the massive datasets needed to support reproducible and statistically significant discoveries. Together, these technical challenges have limited the progress of urinary comprehensive genomic profiling (uCGP) diagnostics.

In this study, we describe the validation of UroAmplitude, a Clinical Laboratory Improvement Amendments (CLIA)-certified NGS-based uCGP test. UroAmplitude provides a sample-to-answer platform with notable technical advances in the collection, storage, extraction, and comprehensive genomic profiling of urine-derived DNA for the noninvasive monitoring of UC. UroAmplitude detects mutations, including single nucleotide variants (SNVs), insertions/deletions (INDELs), loss of heterozygosity, and copy number variants (CNVs), across 60 genes. Here, we use matched tumor and urine specimens to quantify UroAmplitude’s ability to detect tumor mutations in urine DNA. Further, we demonstrate the ability of UroAmplitude to longitudinally monitor residual disease in patients undergoing first-line bladder-sparing treatments including transurethral resection (TUR) and treatment with Bacillus Calmette-Guerin (BCG).

## 2. Methods

### 2.1. Clinical Samples

Across all studies, a total of 341 subjects and 473 samples were prepared for various validations (423 urine specimens for methodology validation and 91 tumor and urine samples for somatic variant quantification) (Appendix A). Cohort recruitment occurred at nine urology practices and collections took place over the course of 57 months. Urology patients, including urologic controls and urothelial carcinoma patients undergoing surgical resection with curative intent were recruited with informed consent for research use at Convergent Genomics. Inclusion criteria for the entire study: 1. 18 years of age or older; 2. Subjects with diagnosis of cancer, healthy controls, and controls with other comorbid conditions; 3. Female and male subjects of any racial or ethnic group; 4. Subject must be competent to provide verbal consent. 

For the prospective somatic variant concordance study (STARD, Appendix A) [36], inclusion required: 1. UC-positive patients where FFPE slides were provided with pathological confirmation of disease; 2. Urine was collected prior to surgical resection. Among 25 concordance analysis evaluated subjects, six were excluded because urine was only collected after resection, and one was excluded due to sequencing sample failure of the paired FFPE specimen (Appendix A). 

Tumor blocks were prepared as FFPE samples following standard fixation and histology practices. FFPE sample sections were stained with haematoxylin and eosin (H&E) and reviewed by a second independent pathologist to assess tumor grading and determine extent of tumor tissue preserved on slides (Appendix A). For DNA extractions, slides with minority percentages of tumor tissue were annotated to highlight tumor and healthy tissue was removed by macrodissection before extraction. All urine samples were collected and preserved in Enhanced Preservation Media (Convergent Genomics, South San Francisco, CA, USA) [37] and transported at room temperature. All samples were stored at −80 °C until processing. 

This study follows the Standards for Reporting Diagnostic Accuracy (STARD) [36] reporting guidelines where applicable. For methods transparency, a STARD methods checklist with applicable page references and author commends is included as a supplement to the paper (Appendix A). 

### 2.2. DNA Purification

FFPE tumor samples were stored at room temperature and protected from light. Samples were extracted using QIAamp DNA FFPE Advanced UNG Kit (QIAGEN, Germantown, MD, USA) according to the manufacturer’s protocol. DNA from EPM-stabilized urine samples were extracted using the UroAmplitude Urine Total DNA Purification Kit (Convergent Genomics, South San Francisco, CA, USA) using a solid-phase reversible immobilization method customized for automation by a Microlab STAR^LET^ (Hamilton, Reno, NV, USA). Briefly, lysis is performed by incubation at 60 °C in the presence of ionic and non-ionic detergents and proteinase K. Following lysis, precipitation is performed using paramagnetic beads, and immobilized nucleic acid is rinsed using an EtOH-based wash. Purified DNA is eluted with low-EDTA TE. Purified DNA concentrations were assessed by Qubit dsDNA HS assay (Invitrogen, Waltham, MA, USA).

### 2.3. Next Generation Sequencing

All NGS sequencing was performed according to the UroAmplitude procedure, a Clinical Laboratory Improvement Amendments (CLIA) validated diagnostic test. Purified tumor and urine DNA was fragmented by sonication (Covaris ME220, Covaris LLC, Woburn, MA, USA) before undergoing library preparation using the KAPA Hyper Prep Kit (Roche, Basel, Switzerland) protocol and is optimized for automation by a Microlab STAR^LET^ (Hamilton). xGen Dual Index adapters are synthesized by IDT and a custom xGen Lockdown Probe panel (Integrated DNA Technology, Coralville, IA, USA) is used for hybridization capture of DNA libraries. The hybridization capture workflow has been optimized for performance on a Nimbus microLAB (Hamilton). Target-enriched libraries are analyzed using a 2100 Bioanalyzer (Agilent, Santa Clara, CA, USA) and diluted for sequencing. Libraries are loaded into a NextSeq 500/550 High Output Reagent Cartridge (v2, 300 cycles) and sequenced on a NextSeq 550 (Illumina, San Diego, CA, USA).

### 2.4. Bioinformatics

Sequenced reads were demultiplexed with Picard Tools [38] (2.18.20), aligned to the human reference genome hg19 with BWA [39] (0.7.17), followed by deduplication with FGBio (0.8.1) [40]. Sequencing tracks are displayed as BAM files with the Integrative Genomics Viewer (IGV) [41]. Variant calling was carried out with VarDict (1.8.3) [42,43], MuTect2 (4.2.3) [44,45], and the custom UroAmplitude algorithm. Optimal somatic variant detection with MuTect2 or VarDict requires either a paired normal specimen or a panel-of-normals. We chose a strategy that leverages a pool of normal urine samples (*n* = 16), as this approach captures technical variability across diverse urine samples and uses the same sequencing approach as future samples of interest. MuTect2 requires a variant calling format (VCF) file consisting of merged variant calls across the panel of normal and was constructed according to the GATK documentation. VarDict was run in paired mode in which the paired normal sample consisted of alignment data from across samples in the panel of normals. 

The custom UroAmplitude SNV calling algorithms were purpose built for somatic mutation detection in urine-derived DNA. Variant calling algorithms were trained on a collection of 1:100 dilutions and the pool of normals described. The algorithms consisted of an isolation forest machine learning approach to identify somatic variants and an empirical modeling approach that uses noise at each location in the capture panel to establish location-specific thresholds for variant calling. A minimum phred score of 30 and a variant allele frequency of 0.15% was used for all variant callers. 

Copy number variants were detected using a *t*-test approach to quantify statistically significant coverage outliers when compared to a distribution of coverages among panel of normal specimens. Copy number neutral loss of heterozygosity was measured by quantifying deviations of the expected VAF (50% or 100%) at common germline SNP locations [46,47]. VarDict (1.8.3) was used to call insertion-deletion events. 

### 2.5. Statistics

The Jarque-Bera test was used to evaluate normality of all parameters. One-way analysis of variance with Bonferonni post hoc correction was used to assess statistical significance among parameters across multiple normally distributed parameters. For continuous variables produced in bioinformatics pipelines we performed Mann–Whitney U-tests. Reported *p*-values < 0.05 were considered statistically significant. True positives (TP), false positives (FP), true negatives (TN), and false negatives (FN) were used to quantify sensitivity = TP/(TP + FN), specificity = TN/(TN + FP), and positive predictive value (PPV) = TP/(TP + FP) for each variant caller. Due to the panel size (>245,000 bp), TNs vastly outnumber both FPs and TPs; therefore, we present PPV to assess algorithm performance instead of specificity (which is >99% in all cases).

## 3. Results

### 3.1. Development of the UroAmplitude Assay for Somatic Mutation Detection in Urine-Derived DNA

UroAmplitude was built to enable the collection, storage, extraction, and comprehensive genomic profiling of urine-derived DNA for the noninvasive monitoring of UC (Figure 1A). To evaluate the UroAmplitude platform, we began by assessing the DNA recovered from clinical urine collections. Here, we asked whether DNA extracted from clinical urine collections would reliably produce sufficient material and would the DNA suffer from degradation. To protect against nuclease activity and other destructive properties of urine, patient samples are stabilized in Enhanced Preservation Media (Convergent Genomics) in the clinic before room temperature transport. Here, we see that stabilized urine samples are protected against nuclease-mediated digestion (observed after only two hours of room temperature incubation) and demonstrate intact, high-molecular-weight DNA when stored at room temperature for up to seven days (Figure 1B and Appendix A). 

Next, urine samples are subjected to a high-throughput extraction procedure that purifies both cell-associated and cell-free DNA. To ensure that common urine contaminants in individuals undergoing a work-up for UC diagnosis do not interfere with preservation and extraction methodology, we investigated extraction yields in a cohort of asymptomatic controls, hematuria patients, and UTI patients. Across all cohorts (423 total samples, Appendix A), we obtained sufficient DNA extraction yields to enable NGS (Figure 1C). Further, we quantified differences in extraction yields among males, females, and individuals with confirmed bladder tumors at time of urine collection. On average, we found that females produce greater DNA yields than men (167 ng to 43 ng), tumor-positive patients out-yield tumor-negative patients (145 ng to 77 ng), and patients with high-grade (HG) disease out-yield patients with low-grade (LG) disease (258 ng to 59 ng) (Figure 1D). Other variables, like smoking history and benign prostatic hyperplasia (BPH) status, did not significantly influence yields (Appendix A). 

We next sought to identify mutations in urine-derived DNA. To target genes that are recurrently mutated in UC, we constructed a custom oligonucleotide hybrid capture panel using public sequencing data on 1586 sequenced bladder tumor specimens [48,49] encompassing a diversity of low- and high-grade tumors and spanning superficial and invasive staging. Locations were chosen to maximize the interrogation of single-nucleotide variants (SNV), insertion-deletions (InDel), and copy number variations (CNV). The final panel includes 60 genes, spanning 245,103 bases of genomic space. 

Early attempts with commercial hybrid capture panels often yielded inconsistent sequencing coverage with wide variance. This resulted in missed mutations in regions with insufficient sequencing information and wasted sequencing at regions with extreme excess. The custom UroAmplitude panel design was optimized to best ensure sequence coverage uniformity across all target regions (Figure 1E,F). To assess panel performance, we compared the sequencing of ten urine DNA samples enriched with both the UroAmplitude and TruSeq Custom (Illumina) hybrid capture panels. In reviewing coverage depth across individual genes and exons, we found that the coverage uniformity across large exons was significantly improved with UroAmplitude; longer exons captured with TruSeq often demonstrated extreme variability, with individual exons having 10- to 20-fold variation of depth across their length (Figure 1E). Across small exons, we observed uniform coverage within the exon using either Illumina TruSeq or UroAmplitude panels, but TruSeq produced greater depth-of-coverage variation from exon to exon (Figure 1E). 

Next, we measured the coverage uniformity of the entire hybrid capture panel by comparing the distribution of relative coverage for the UroAmplitude and TruSeq panels. (Relative coverage is defined as the coverage at a specific location normalized to the median coverage of all locations in the panel.) Here, we found TruSeq probes produced higher coverage variability, with a standard deviation 2.1 times that of UroAmplitude. At the extremes of the distribution (shaded in red), we found TruSeq produced 5.8 times more locations with high variability than UroAmplitude (Figure 1F). Together, these data show that the hybrid capture uniformity achieved by the UroAmplitude custom panel produces more consistent sequencing performance.

To detect somatic SNVs in urine-derived DNA, we set about testing the widely-used variant callers VarDict (1.8.3) [42,43] and MuTect2 (4.2.3) [44,45]. As our diagnostic paradigm operates in the absence of a paired normal tissue specimen (e.g., blood or saliva), the variant caller must independently account for technical noise and successfully distinguish somatic from germline SNVs while maintaining high sensitivity and positive predictive value (PPV). Following the recommended protocol for non-paired modes [42,44], a panel of normal samples (*n* = 16) was assembled to serve as a noise model during variant calling with both VarDict and MuTect2. These variant callers were then used to call spike-in variants created by diluting one reference sample into the background of another (Coriell Institute, NIGMS reference repository). We explored dilutions of 1:10 and 1:50, in triplicate, to produce simulated somatic variants (diluted germline variants) with expected variant allele frequencies (VAFs) of 10%, 5%, 2%, and 1%. MuTect2 demonstrated 57% sensitivity in pure sample but achieved only 0.5% and 4.1% sensitivity in 1:10 and 1:50 dilutions, respectively (Figure 1G). In contrast, VarDict performed with a sensitivity above 90% across all dilutions (Figure 1G); however, this performance came at the expense of a high number of false positives, as 1:10 and 1:50 dilutions achieved 40% and 17.4% PPV, respectively (Figure 1H).

To overcome the challenge of low PPV when calling somatic SNVs, while maintaining high sensitivity, we developed an ensemble variant-calling approach purpose-built for urine-derived DNA. Model parameters were optimized using a set of reference and replicate samples to account for technical and biological variability. The ensemble model was trained to a sensitivity of 91% and PPV of 94%. The model was then validated on a set of held-out reference samples and was observed to have a sensitivity above 97% with a PPV above 80% for variants with 5% and 1% allele frequency (Figure 1G,H, Appendix A).

### 3.2. Comprehensive Genomic Profiling of UC: Comparison of Tumor and Urine

To assess the ability of UroAmplitude to comprehensively profile the genomic characteristics of UC, we next investigated a collection of tumor specimens for which urine was available either at time of diagnosis or following resection (Appendix A). Across the 23 patients whose tumor and urine samples were profiled (67 total samples), *TERT* promoter mutations were the most common. We observed *TERT* promoter mutations in 61.7% of UC-positive samples, with a moderate enrichment in HG disease (Figure 2A,C). However, in cystoscopy-negative urine samples, *TERT* promoter mutations were rare (4.8% of samples). *PLEKHS1* promoter and *FGRG3* missense mutations were the next most frequently observed variants, at 38.3% and 36.2%, respectively, followed by *TP53* mutations at 31.9%. Consistent with previous reports [48], *FGFR3* and *TP53* mutations are the most powerful indicators of cancer grade, with *FGFR3* mutations enriched in LG and *TP53* mutations enriched in HG tumors (Figure 2A,C). Genes responsible for encoding critical effectors and regulators of chromatin remodeling were also frequently mutated and enriched for deletion events. The most frequently mutated among these was *KMT2D* (31.9% of disease-positive samples), followed by *ARID1A*, *KDM6A*, *CREBBP*, *KMT2C*, and *KMT2A* (12.8%). In general, SNV events in epigenetic regulators were found to be evenly represented in both HG and LG disease (Figure 2A,D). Mutations in tumor suppressors (such as *TP53* and *STAG2*) and epigenetic regulators were occasionally observed in cystoscopy-negative urine samples of patients with a prior history of bladder cancer (2.3× enrichment in tumor-positive versus post resection negative cystoscopy samples); whereas oncogenes were rarely observed in post resection samples. For example, *FGFR3* was enriched greater than 8x in tumor-positive samples, while *PIK3CA* mutation and *FGF3* amplification were never observed in post resection, negative surveillance cystoscopy samples (Figure 2A,B).

We also assessed genome alterations by measuring both CNV and copy-number-neutral loss of heterozygosity (LOH) across the entire panel. Significant CNV activity was not observed in cystoscopy-negative UC surveillance urine samples, while HG tumor samples demonstrated the most substantial CNV alterations among disease-positive samples (Figure 2B). *SPAG1* copy number gain was the most frequently observed CNV event and was heavily enriched in HG disease (Figure 2A,E). *CDKN1A* gain was the next most prevalent CNV event and was observed evenly in HG and LG disease. *CDKN2A* loss and *FGF3*, *PAIP1*, and *SOX4* gains were also frequently observed, but only in HG disease (Figure 2E). Finally, we found that LOH events were enriched in HG patients compared to LG patients (*p*
*=* 0.0367, Mann–Whitney U), with twice the number of events in HG samples on average (Figure 2B,F). 

In summary, HG tumors and matching urine samples were enriched in *TP53* mutations, LOH events, and genes demonstrating CNV, whereas LG disease was enriched in *FGFR3* mutations and rarely demonstrated significant CNV activity, especially in urine collections.

### 3.3. Somatic Variant Concordance: Comparing Mutations in Urine to Matched Tumor

To clinically validate UroAmplitude SNV detection, we applied the assay to 18 urine specimens from UC patients, taken prior to tumor resection, for which matched tumor sequencing was available (Appendix A). We next compared the mutations identified in the matched tumor and urine samples to access concordance (Appendix A). Tumors were sequenced to an average depth of 462 genomes, which supported confident variant calling down to the sub-clonal VAF cutoff of 10% established by Gerstung and colleagues [50]. Among SNVs detected, we observed that a median of 91.7% of mutations found in tumor DNA were also found in the urine DNA, while a median of 57.6% of mutations found in urine DNA were also found in tumor DNA (Figure 3A). We next assessed the respective VAFs of concordant mutations found in both tumor and urine. Here, we observed that subjects with HG disease consistently demonstrated higher average VAFs in the urine than those with LG disease (*p* = 0.0046, Mann–Whitney U). HG tumor VAFs largely recapitulated the VAFs observed in the urine above 10%, approximating a 1:1 trendline, whereas LG tumor mutations tended to exhibit more diluted frequencies in the urine (Figure 3B). Finally, we investigated mutations that were observed in the urine but not found in the tumor. In general, urine-exclusive events had low VAFs, with 72.5% of variants having VAFs below 1.0%. We also found that urine-exclusive events were 64% more likely to be found in HG samples than LG samples, and at higher average VAFs, consistent with urine-exclusive events having potential biological and clinical relevance (Figure 3C).

We next assessed tumor-concordant and urine-exclusive SNV events found in the urine of individual patients. Across three unique patient cases (Figure 4A–C), we analyzed matched tumor-urine sequencing from urine samples collected at the time of diagnosis. In each case, UroAmplitude identified UC-associated mutations with 100% concordance (tumor mutations found in urine). In the case of Patient 42, UroAmplitude accurately profiled the tumor from urine samples collected during a check-up for hematuria and again when the patient returned for cystoscopy follow-up (Figure 4B). In all cases, low VAF urine-exclusive events were identified that could be distinguished from the defining genomic variants of the primary tumor.

Finally, we reviewed SNVs in two patients with HG invasive cancer diagnoses which also had high numbers of urine-exclusive events (Figure 4D,E). In both cases, the mutational signatures of the tumor specimens were evident in the UroAmplitude profiling of the matched urine samples, and we observed 100% and 83.3% concordance for patients 64 and 70, respectively. We also identified numerous urine-exclusive, secondary mutations (ten and five, respectively) that were present with VAFs an order of magnitude lower than the core mutations of the tumor (Figure 4D,E). 

Altogether, these data illustrate that the UroAmplitude platform can establish the mutational profile of a patient’s tumor from a urine collection taken prior to surgery.

### 3.4. UroAmplitude Case Studies Monitoring Residual Disease following TURBT

In addition to urine collections taken prior to tumor resection, we sequenced longitudinal urine collections taken immediately following resection, or over the course of post-surgery surveillance. Using these longitudinal collections, we asked whether we could detect disease recurrence following resection.

As expected, we saw that patients with HG disease were most likely to demonstrate residual mutation signatures from the primary tumor in the urine following resection. Patients 31 and 46 were particularly clear examples of residual primary disease following surgery. Sequencing of the primary tumor from Patient 31 revealed four core mutations, with three present above 40% VAF: *ARID1A*, *TERT*, and *TP53* (Figure 5A and Appendix A). Three days following resection of this tumor, urine was collected during routine follow-up, and all three high-VAF mutations remained present in the urine at VAF greater than 5%, indicating the presence of residual primary disease. One month later, after a restaging surgery was performed, pathology reported the resected tissue negative for tumor. A subsequent urine collection revealed no remaining evidence of the primary tumor, suggesting the restaging surgery was successful (even though tumor was not observed in the tissue reviewed by pathology). A surveillance urine collection 7 months after the first resection again showed no evidence of the primary tumor, and the patient has remained cystoscopically negative for 21 months. 

Sequencing of the primary tumor from Patient 46 revealed six mutations with VAFs between 30–80%. Using UroAmplitude we observed these same six mutations in a urine collection following initial resection of this tumor (Figure 5B and Appendix A). Consistent with persistent urine mutations, pathology from the restaging surgery found evidence of HG disease and the patient underwent radiation and chemotherapy. At 6.5 months, a follow-up cystoscopy was negative, but urine collected at the time continued to measure dominant tumor mutations, with both highest-frequency mutations (*STAG2* and *KMT2D*) remaining. At 13 months, disease progression was observed by CT, with metastatic tumor spread to the abdomen and spine.

These findings illustrate UroAmplitude’s ability to detect residual disease in the urine of patients following resection of their primary tumor. When combined with time-of-diagnosis testing (Figure 4), UroAmplitude has the capacity to characterize and surveil a patient’s disease strictly from urine collections. In Patient 22, we present a case where urinary profiling is performed both before and after resection (Figure 6). Here, five mutations with variant allele frequencies around 10% are measured in urine collected at the time of cystoscopy. The five mutations include *FGFR3*, *TERT*, and three epigenetic regulators, and together are indicative of a LG tumor. Following resection, pathology confirmed the tumor as LG and UroAmplitude sequencing of the tumor found 100% mutation concordance between urine and tumor. A follow-up collection, taken 13 days after surgery, showed no evidence of any of the five defining tumor mutations, suggesting the surgery was successful in removing the entirety of the tumor. Urine collections over the following seven months showed no evidence of residual/recurrent disease. Surveillance cystoscopy has continued for this patient up to 19 months, and they remain tumor negative.

### 3.5. Intravesical Therapy Response Case Studies

Next, we tested the performance of UroAmplitude for detecting residual disease burden in patients receiving intravesical therapy (IVT) following resection of an HG tumor, using three patients as case studies. In these case studies we explore UroAmplitude testing utility when a urine sample is available either before resection or prior to BCG induction, but where a paired tumor specimen was not available. For Patient 6 (Figure 7A), urine was collected before and after surgery and throughout the course of Bacillus Calmette-Guerin (BCG) treatment. UroAmplitude identified *TERT*, *PLEKHS1*, *FOXQ1*, *KMT2C*, and *ERBB2* somatic events in urine collected prior to resection of an HG T1 multifocal tumor. In a subsequent urine sample taken following resection but prior to the first instillation of BCG, the same constellation of mutations was present, albeit at reduced VAFs, indicating incomplete resection (Figure 7A). Over the course of BCG instillation, this core set of mutations was gradually reduced, with late clonal [50] mutations no longer detected after the third instillation, and early clonal mutations no longer detectable by the final instillation. A *KMT2C* mutation—present at the time of first collection at 2.5% VAF—originally appeared to be a sub-clonal mutation of the primary tumor, but unlike all other core mutations, its allele frequency was not reduced following resection. In addition, this mutation persists throughout the course of IVT and grows in allele frequency.

Patient 10 (Figure 7B) presents an example of a tumor-naïve paradigm in which surveillance specimens were acquired only after surgery for HG Ta/CIS disease and initiation of IVT. UroAmplitude measured *TERT* and *FOXA1* mutations at 8.4% and 4.3% VAF, respectively, prior to the second BCG instillation. These mutations were significantly attenuated over the course of treatment, decreasing to 0.36% (*TERT*) and 0.26% (*FOXA1*) and ultimately zero at the time of a negative cystoscopy following the BCG course. Three months later, at a surveillance follow-up, UroAmplitude again detected the *TERT/FOXA1* mutations, indicating tumor recurrence that was confirmed by cystoscopy. A subsequent TUR was performed and confirmed the continued presence HG Ta/CIS disease. 

Lastly, we examined Patient 13, who had been diagnosed with HG multifocal CIS and for whom the first urine was collected after resection (Figure 8). During the primary TUR, foci of CIS were removed from the posterior, left, and right lateral walls of the bladder as well as from the prostatic urethra. UroAmplitude surveillance of Patient 13 began on the first day of BCG induction, where we identified a mutation burden characterized by high-VAF mutations in *TERT* (52.3%), *PLEKHS1* (34.1%) and *TP53* (32.5%). Over the 6-week course of the first BCG induction, the mutational burden was substantially reduced, leaving core mutations at just ~1% VAF. Follow-up cystoscopy at 15 weeks was negative and cytology was atypical/inconclusive. 

Maintenance BCG began at 21 weeks. UroAmplitude showed an increase in primary tumor mutation frequency since completion of BCG induction, and a slight reduction by the conclusion of the maintenance course. At 29 weeks, cystoscopy and cytology were negative. At 37 weeks, a second BCG maintenance course began and UroAmplitude again detected a substantial increase in mutation frequency. Core mutations in *TERT* and *TP53* returned to ~30% VAF, but post-maintenance cystoscopy and cytology remained negative. During a surveillance visit at 55 weeks, cystoscopy was again negative, but urine cytology was suspicious; UroAmplitude mutations were now observed as high as 40% VAF (Figure 8).

In response to suspicious cytology, at 62 weeks Patient 13 underwent blue-light-assisted cystoscopy that was not able to visualize tumor. Random mapping biopsy was performed that confirmed HG CIS on the posterior wall of the bladder and “atypical, favor reactive” tissue from the prostatic urethra. Following surgery, Patient 13 began a second BCG induction while UroAmplitude analysis measured a further increase in mutational burden, as high as 70% VAF. Sample analysis following the second BCG induction showed no significant reduction of mutant allele frequency, and a final urine collection, taken at 82 weeks, showed core tumor mutations persisting at high allele frequency. Cystoscopy at this time remained negative, but cytology returned a definitive positive. Patient 13 was then referred to a Comprehensive Cancer Center for continued treatment, and blue-light-cystoscopy-guided TUR found a HG T4a lesion of the prostatic urethra, with extensive invasion into the prostate (Figure 8). Together, these longitudinal studies illustrate the ability of UroAmplitude to act as a noninvasive companion diagnostic to IVT by enabling assessment of BCG response and treatment durability.

## 4. Discussion

Due to the invasiveness, morbidity, and limited performance of cystoscopy—coupled with urinary cytology’s lack of clinically performant sensitivity—there is a clear unmet need for urine-based tools for diagnosis, surveillance, and screening of high-risk UC populations. Biomarker technologies introduced over the past decade have brought about increased sensitivity of UC detection compared to urine cytology but at the expense of decreased specificity, preventing their adoption in international guideline recommendations [17,51]. In many cases, biomarker assays are either contraindicated or unable to provide confident results in common urologic scenarios, including recent instrumentation/catheterization, presence of UTI, and during IVT. Other biomarker testing algorithms rely heavily on the input of already established clinical risk factors provided by the ordering physician, failing to provide significant unique insight. These factors limit these biomarkers’ real-world performance, and together with an absence of substantive clinical benefit studies, have kept these technologies from achieving widespread clinical adoption.

Here, we introduce UroAmplitude, a non-invasive, urine-based NGS approach to comprehensively profile the genomic aberrations underlying UC. UC is a heterogeneous disease state in which individuals harbor diverse drivers of oncogenesis (e.g., driver mutations, copy number alterations, epigenetic dysregulation) that cannot be captured in a low-dimensional biomarker assay. We validate UroAmplitude’s ability to directly measure these individualized root causes by genotyping primary tumors and demonstrating a >90% concordance rate of tumor tissue mutations observed in urine specimens. In contrast to biomarker-based assays that focus solely on disease presence/absence, we illustrate how UroAmplitude provides actionable clinical insights by enabling longitudinal genomic profiling (whether tumor-informed or tumor-naïve) to support key clinical decision points faced by urologists: post-TUR assessment, recurrence monitoring, and treatment escalation in individuals where intravesical therapy is failing to reduce residual disease burden.

In this study, we detail the technological advances made to turn the promise of uCGP into clinical-grade testing reality. Many key improvements came from customizing general purpose technologies to our specialized objective. For example, VarDict and MuTect are well characterized general use variant callers that are versatile and serve broad applications. In contrast, the UroAmplitude variant caller was trained using ultra-deep sequencing of UC-specific genome locations, with urine derived DNA, and UroAmplitude-specific sequencing methodology. As such, it is not surprising that UroAmplitude’s variant caller has a significant performance edge on these general-purpose variant callers in the context of this study.

Though we demonstrate the considerable potential of UroAmplitude, our approach requires continued work. The matched-tumor cohort presented here is modest in size, and we test for more genes than we have analyzed samples. We do observe frequently mutated genes, such as *TERT*, *FGFR3*, and *TP53*, at rates we would expect based on the literature [48], further evidence of the accuracy and validity of UroAmplitude; but with the study’s current size, we are not statistically powered to assess the prevalence of less frequently mutated genes. As expected, CNV detection was muted in urine samples compared to pure tumor; this was particularly apparent in the low rate of copy number loss events observed in urine (such as *CDKN2A*) compared to that in the literature [52]. Because variant calling is tuned for maximal PPV, copy number events are underrepresented in tumors shedding a low tumor fraction into the urine. Despite a reduced sensitivity for low-level gene loss, we do find a strong correlation between CNV activity and both the pathologic grade and stage of a patient’s cancer. HG and invasive tumors are notably enriched in urinary CNV activity, supporting the prognostic value of the approach. This study provides validation of UroAmplitude’s variant calling in a modest prospective cohort, but disease classification and risk assessment will need further retrospective and prospective validation across larger cohorts. Nevertheless, the clinical case studies presented here support UroAmplitude as a sensitive means to diagnose UC and predict long term outcomes. Finally, UroAmplitude has not yet been validated against rare urothelial and non-urothelial malignancies. Adenocarcinoma and paraganglioma of the bladder, as well as rare UC subtypes are more likely to present with mutations outside of the UroAmplitude 60 genes panel and are therefore more likely to evade detection.

### 4.1. Minimal Residual Disease

By studying the mutational dynamics following an initial resection, we demonstrate UroAmplitude’s ability to support clinical assessment of TUR efficacy. In two case studies of initial HG T1 and T2 diagnoses (Figure 5), each of these patients was immediately scheduled for a follow-up restaging surgery without any clinical insight regarding the effectiveness of their initial surgery. In each case, the patient’s defining tumor mutations were reduced in signal intensity but persisted in the urine following initial surgery, suggesting residual disease endured in the margin of the initial biopsy site. These case studies illustrate how UroAmplitude may inform a physician of residual tumor in the bladder following surgery, and reinforces the value of surgical re-resection for high-risk patients. These cases also reveal an additional potential insight, namely that monitoring reduction of specific mutation allele frequencies over time and in response to surgery can determine whether a patient has residual margin at the original resection site or a secondary independent/multifocal tumor. Such insights can help guide clinical decision-making: in one scenario, where mutations are reduced by surgery but not eliminated, a secondary resection at the initial surgery site is probably beneficial. In another scenario, where high allele frequency mutations persist and are not reduced by a surgical intervention, exploratory mapping biopsy or enhanced blue-light cystoscopy may be more appropriate. In both presented cases, the successful clearance or persistence of tumor mutations after restaging surgery might have helped to guide physician-patient decision-making around the appropriateness of bladder-sparing approaches verses radical cystectomy [53,54].

### 4.2. Monitoring Response to Intravesical Therapy and Early Identification of Therapeutic Resistance

Although intravesical therapy (IVT) is the standard of care following surgical resection of HG tumors, currently there are no diagnostic tests to monitor treatment efficacy. Many biomarker-based assays are either contraindicated or challenged by treatment-related inflammation and are unable to provide meaningful results when used near BCG treatment. Rates of atypical/indeterminate urine cytology can be as high as 70–90% in the months following BCG [55]. Further, interpretation of cystoscopy is often difficult in the setting of post-treatment erythema, which can be challenging to distinguish from recurrent malignancy [56]. Thus, there is a pressing unmet need to provide clinicians the ability to assess treatment response to IVT interventions.

Here, we demonstrate UroAmplitude’s ability to track a patient’s response to IVT in real-time, documenting disease evolution at each BCG instillation. In all three case studies presented, patients have substantial disease burden after surgery but respond favorably to induction [57,58]. Patient 6 (Figure 7A) is a best-case scenario, as they responded robustly to induction and remained disease-free for the duration of surveillance. Patient 10 (Figure 7B) initially responds, but they do not see appreciable change in tumor burden until the end of their induction cycle, and recurrence is detected during surveillance three months later. Patient 10’s late-cycle response raises the question of whether they could have benefited from more than the standard six cycles of induction. Finally, Patient 8 (Figure 8) provides a perspective where longitudinal urine specimens were available across multiple courses of BCG induction and maintenance. In contrast to the first two cases, this patient showed minimal response to multiple treatments. UroAmplitude identifies a continuous and significant tumor burden throughout treatment and surveillance. Their tumor is defined by a TP53 mutation that carries a poor prognosis with increased risk of resistance to neoadjuvant chemotherapy, tumor invasion, and metastasis [48,59]. Despite careful implementation of standard of care, it takes 20 months for this patient to receive a referral to a uro-oncology specialist, where they are ultimately diagnosed with an extensively invasive T4a tumor. Though this case-study is observational and retrospective, the high-risk features identified by UroAmplitude early on and throughout this patient’s course of treatment could have provided more than a year of lead-time to identity the recurrent tumor and possibly prevent its invasive and potentially life-threatening progression. In whole, these case studies illustrate the ability of UroAmplitude to monitor response to intravesical therapy in both tumor-informed and tumor-naïve longitudinal surveillance.

### 4.3. Clinical Utility Today and Expanding in the Future

The insights provided by UroAmplitude are actionable today and may inform surgical planning, risk stratification, surveillance intensity, and the need for blue-light cystoscopy and mapping biopsies. Differentiation of patients who do not respond to BCG or who have residual muscle-invasive disease may identify strong candidates for cystectomy or FDA-approved therapies (valrubicin, pembrolizumab, atezolizumab, or erdafitinib); decreasing the time to surgical and targeted therapeutic interventions can save lives [60]. Informed by future interventional studies, UroAmplitude may also allow for patient-tailored dosing of IVT or accelerated re-induction. The actionability of these insights will be further enhanced through application of machine learning algorithms to define the likelihood of a positive cystoscopy, the likelihood of having pathologically HG cancer identified on biopsy, and the risk of tumor invasion, all based on urinary genomic profiles. Machine learning also has the potential to provide predictive insights into recurrence risk and timing to recurrence. To realize this potential, larger studies will need to be performed. A common challenge in genomic studies is the vast scale of measurement and the need for large clinically diverse cohorts to identify reproducible correlations between somatic mutations and patient outcomes.

The benefits of urinary comprehensive genomic profiling will increase with time and larger clinical studies. For example, the identification in this study of low allele frequency bladder-clonality-associated mutations (see patients 6, 64, 70) are an intriguing risk factor which may predict long-term recurrence or therapeutic resistance [61]. In fact, in the initial diagnosis setting, recent research has shown that at least one type of clonality mutation can precede (and likely predict with high specificity) UC as early as 10 years in advance of clinical diagnosis [62]. However, debate remains over how to best manage patients whose disease (or predicted disease) is months or years from being clinically visible by cystoscopy.

Today, a rich pipeline of therapeutic development in urologic oncology means that the gene-level insights contained within UroAmplitude will become progressively more applicable and actionable over time. We believe a future exists wherein patients with high-risk pre-malignant mutations are identified years before the development of invasive tumors. Timely definition of this risk will provide ample opportunity for lifestyle modification and chemo-preventative intervention, similar to programs implemented today in hereditary cancer syndromes and breast cancer prevention [63].

## 5. Conclusions

We have developed a sample-to-answer platform that enables the collection, room temperature storage, DNA extraction, and comprehensive genomic profiling of urine-derived DNA for the noninvasive monitoring of UC. In this report, we have demonstrated the ability of UroAmplitude to detect somatic variants in both early and late-stage UC and provide insights to aid critical clinical decisions across multiple disease scenarios and case studies.

## Figures and Tables

**Figure 1 jcm-11-05827-f001:**
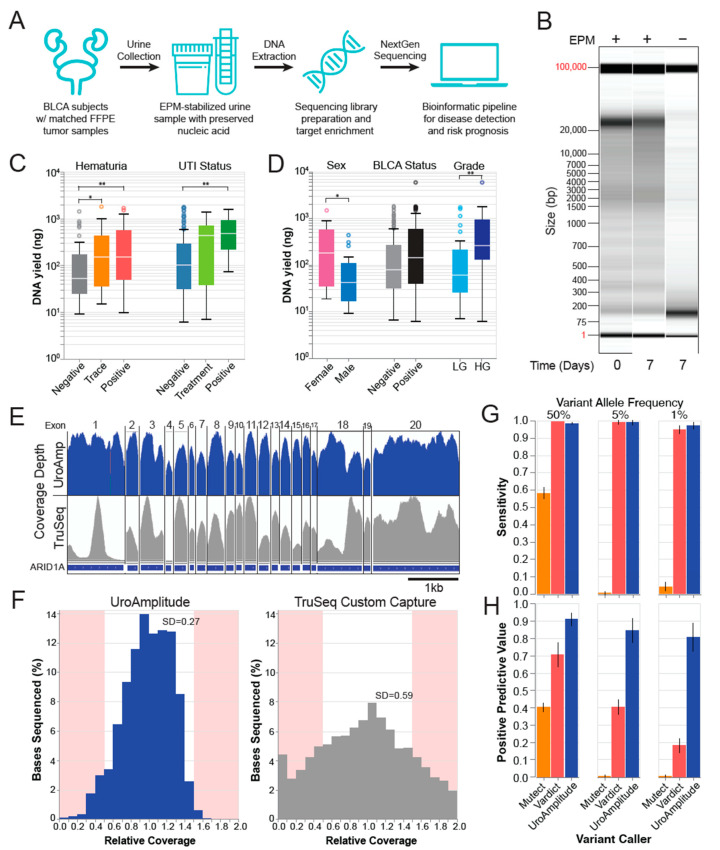
**UroAmplitude workflow schematic and technical validation.** (**A**) Urine samples are preserved in EPM at collection and shipped to Convergent Genomics laboratories where samples undergo automation-assisted DNA extraction, library preparation, target enrichment, and NextGen sequencing. A custom bioinformatics pipeline detects SNVs and CNVs and predicts disease status and risk. (**B**) Fragment Analyzer (FA) “gel view” of urine DNA purified immediately or after 7 days of RT incubation unbuffered or EPM stabilized. (**C**,**D**) Box plots of median extraction yields from 4ml urine aliquots for key urology clinic demographics: sex (*n* = 21), disease status (*n* = 272) and grade (*n* = 73), hematuria status (*n* = 92), and urinary tract infection (UTI) status (*n* = 139).* *p* < 0.05, ** *p* < 0.01. (**E**) Representative IGV tracks of gene-level sequencing coverage following UroAmplitude or Illumina TruSeq capture. (**F**) Relative coverage (coverage at one location normalized to the median coverage across all locations) distributions for UroAmplitude and Illumina TruSeq hybrid capture methodologies. Sites outside 50% variance are highlighted. (**G**,**H**) Blinded, independent validation of technical sensitivity and Positive Predictive Value (PPV) for simulated somatic variants with heterozygous Variant Allele Frequencies (VAF) of 50%, 5%, and 1%. Three different variant callers are compared (MuTect, Vardict, and UroAmplitude).

**Figure 2 jcm-11-05827-f002:**
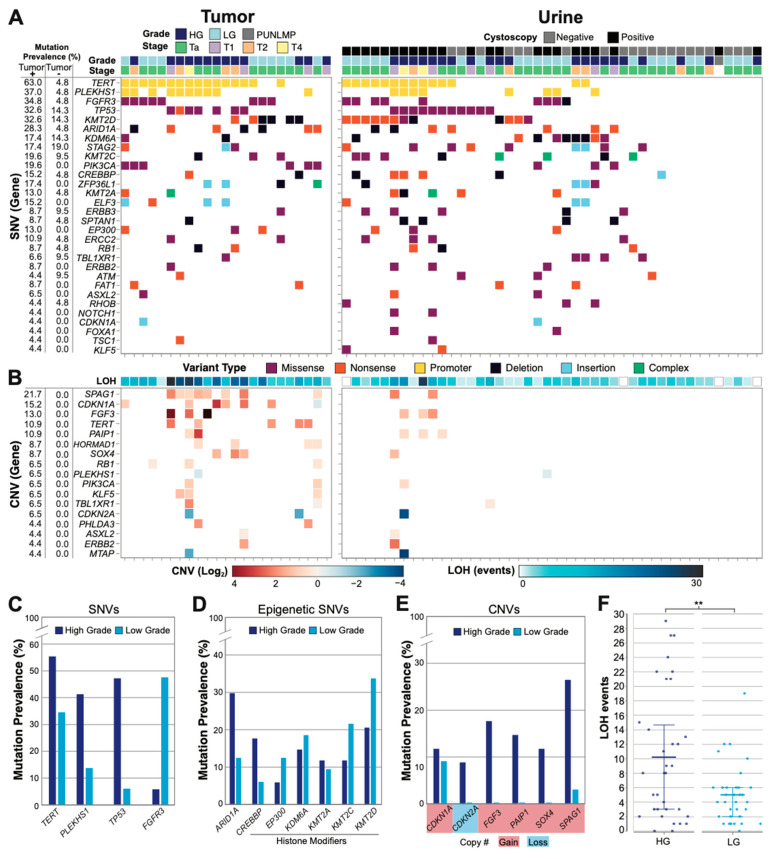
**Genomic profile of the urothelial carcinoma cohort.** (**A**) Tumor (left, *n* = 23) and urine (right, *n* = 44) SNVs are illustrated for each patient sample with pathologic correlates (one column per patient). All genes mutated in >2.5% of tumor positive samples are shown. For urine samples, the cystoscopy status is color coded. Cystoscopy positive and negative urines are collected before and after tumor resection, respectively. (**B**) Significant CNV events (log2 fold change) and copy-neutral loss of heterozygosity (LOH) (total events) are shown. All CNV events present in >2.5% of tumor positive samples are shown. (**C**) Mutation prevalence in HG and LG disease are shown for the most frequently identified genes, genes that correlate with grade, and (**D**) genes responsible for epigenetic regulatory mechanisms. (**E**) Mutation prevalence of genes with most frequently observed CNVs, with gains or losses highlighted. (**F**) LOH events are plotted for HG (*n* = 34) and LG (*n* = 32) disease with mean and 25–75-percentiles highlighted (** *p* = 0.0367, Mann–Whitney U).

**Figure 3 jcm-11-05827-f003:**
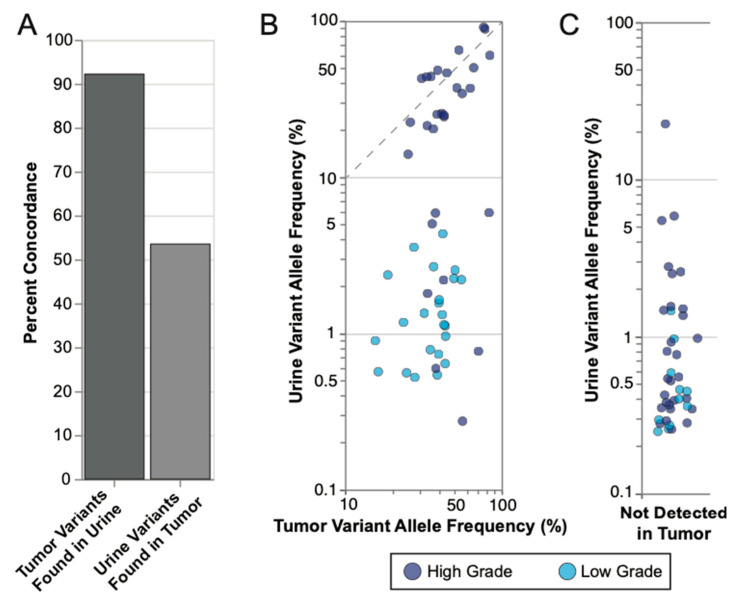
**UroAmplitude mutation concordance between tumor and matched urine.** (**A**) Median concordance of mutations measured in matched tumor (*n* = 18) and urine samples are shown for tumor mutations found in urine and urine mutations found in tumor. (**B**) Concordant mutations were plotted according to their respective VAFs in tumor and urine with high grade (HG) samples in dark blue and low grade (LG) samples in light blue. Significance of mean VAFs was assessed using Mann–Whitney U (*p* = 0.0046). The dotted line highlights a 1:1 VAF ratio for Urine and Tumor variants above 10%. (**C**) Urine exclusive events (mutations not found in the tumor) were plotted according to the VAFs with HG and LG samples highlighted.

**Figure 4 jcm-11-05827-f004:**
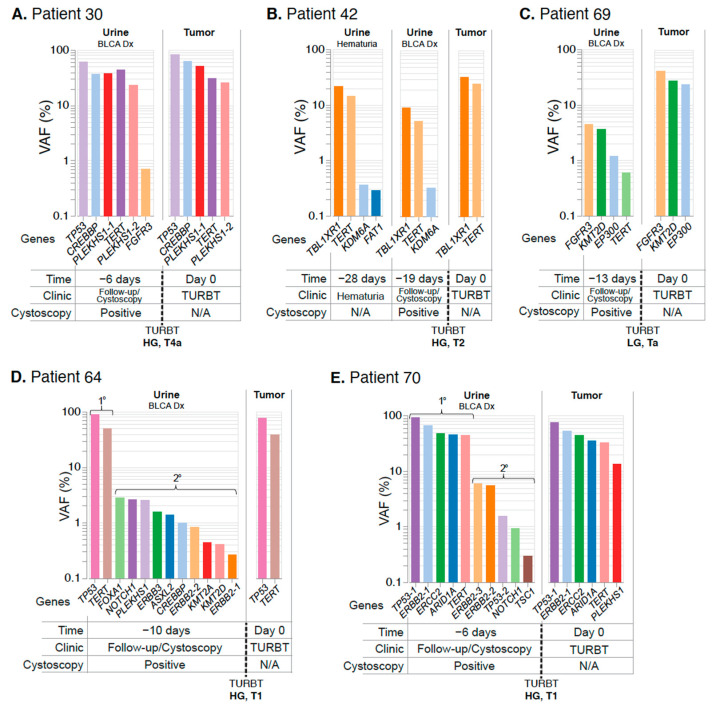
**UroAmplitude urine analysis at the time of diagnosis establishes a comprehensive genomic profile of the tumor.** The mutation profile of matched urine and tumor are grouped chronologically to demonstrate the somatic mutation concordance between urine samples at diagnosis and resected tumor. Tumor mutations are color-coded to facilitate identification across collections. When multiple mutations are identified in the same gene (ex. PLEKHS1 in Patient 30 and ERBB2 in Patient 70), each mutation is distinguished by color and given a numeric suffix. (**A**–**C**) demonstrate high tumor-in-urine concordance with few urine-exclusive events. (**D**,**E**) demonstrate high tumor-in-urine concordance with high numbers of urine-exclusive events (denoted as secondary). Abbreviations: primary (1°), secondary (2°), diagnosis (Dx), transurethral resection of bladder tumor (TURBT).

**Figure 5 jcm-11-05827-f005:**
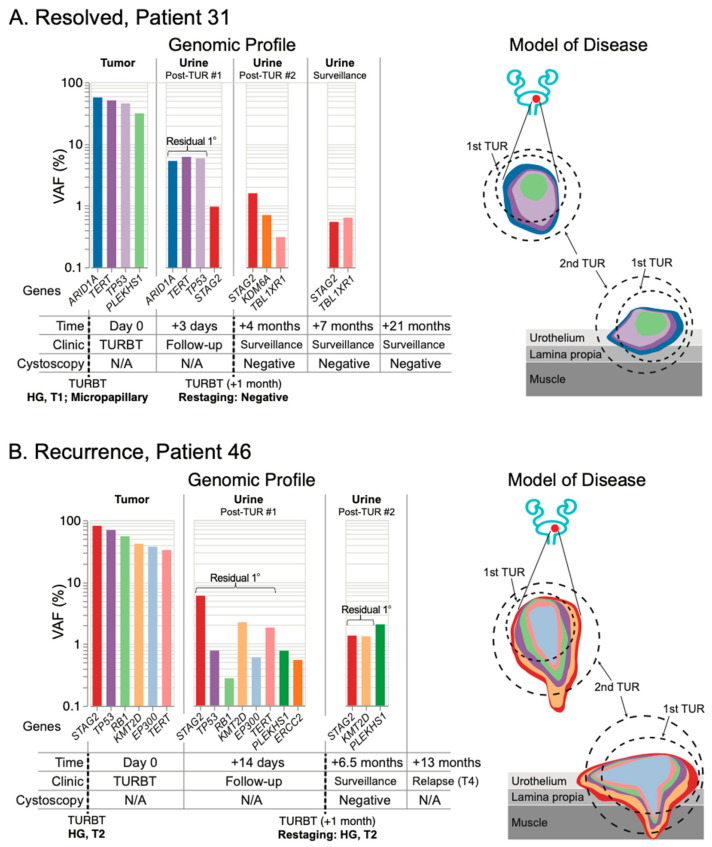
**Post-resection mutation profiling reveals residual primary disease in surveillance urine collections.** The mutation profile of matched tumor and urine samples are grouped chronologically to demonstrate genomic changes over the course of treatment. Tumor mutations are color-coded to allow surveillance across longitudinal collections, with residual primary (1°) tumor mutations present in the urine highlighted. Disease modeling incorporates UroAmplitude genomic features (mutation color-coding) with clinical treatment and pathology. (**A**) Patient 31 demonstrates residual primary tumor mutations after the initial transurethral resection (TUR), but successful elimination of these mutations following restaging of the primary site. The patient has no evidence of visibly apparent recurrence out to 21 months of surveillance cystoscopy. In contrast, (**B**) Patient 46 demonstrates persistent primary tumor mutations both after the primary TUR and after restaging. The patient experiences metastatic progression detected 13 months following initial TUR.

**Figure 6 jcm-11-05827-f006:**
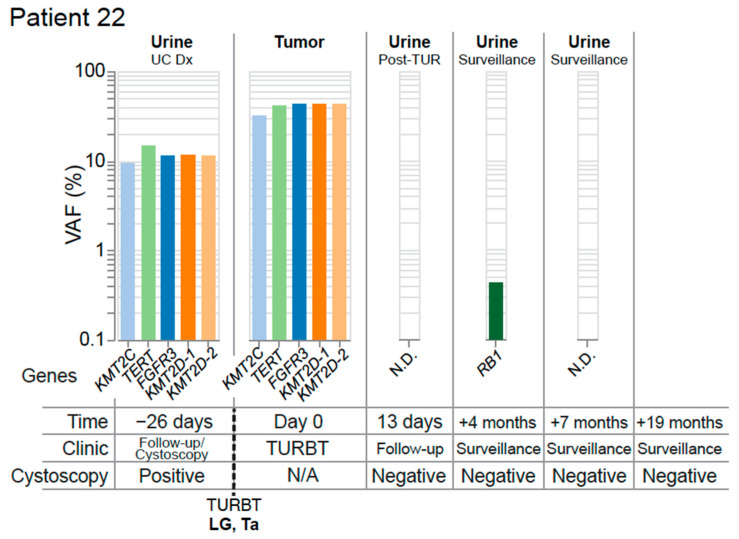
**Complete resection with optimal molecular surveillance.** The mutation profile of matched tumor and urine samples from Patient 22 are grouped chronologically to demonstrate genomic changes over the course of treatment. Tumor mutations are color-coded to facilitate identification across longitudinal collections. Multiple mutations are identified in the gene KMT2D, and are distinguished by color and numbered. Abbreviations: none detected (N.D.), diagnosis (Dx).

**Figure 7 jcm-11-05827-f007:**
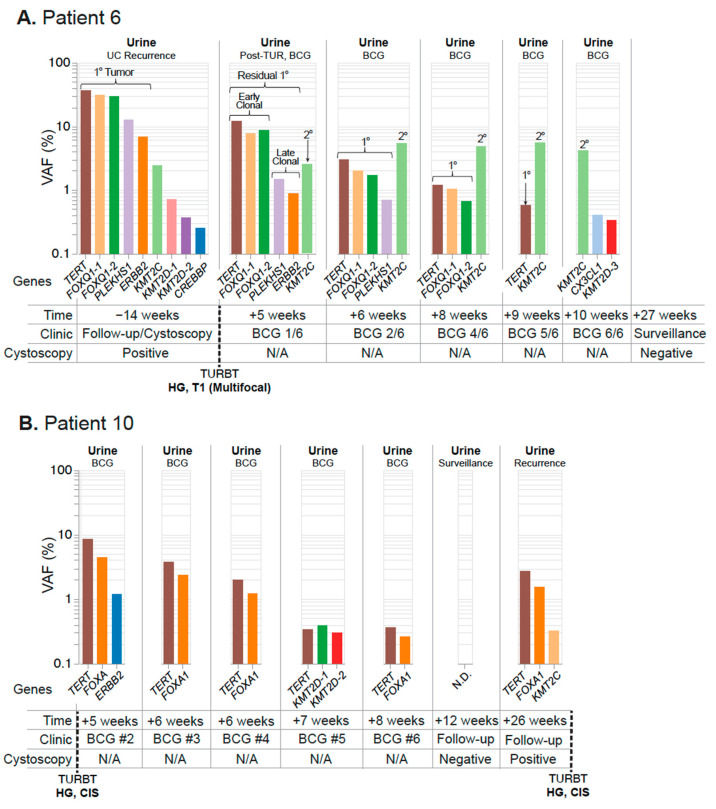
**Residual disease surveillance during intravesical therapy.** Longitudinal mutation profiles in urine samples taken over the course of BCG instillation. Mutations are grouped chronologically to demonstrate genomic changes over the course of treatment. Tumor mutations are color-coded to allow surveillance across longitudinal collections, with residual primary tumor mutations present in the urine highlighted. (**A**) Patient 6 surveillance begins at the time of a positive cystoscopy and continues after TUR and throughout the course of BCG instillation. Putative early and late-clonal mutations of the primary (1°) tumor are highlighted, as well as a secondary (2°) site. (**B**) Patient 10 surveillance begins after their first BCG instillation and continues throughout the course of treatment until tumor recurrence at 26 weeks. None detected (N.D.).

**Figure 8 jcm-11-05827-f008:**
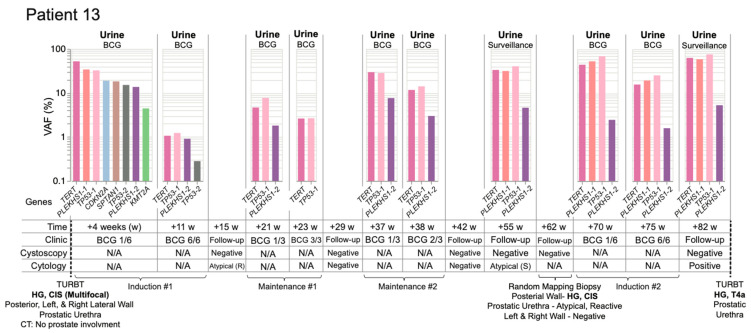
**UroAmplitude predicts persistent disease and invasive progression with a 20-month lead-time.** Longitudinal mutation profiles in urine samples taken over the course of BCG instillation. Mutations are grouped chronologically to demonstrate genomic changes over the course of treatment. Tumor mutations are color-coded to allow surveillance across longitudinal collections. Patient 13 surveillance begins at the time of first BCG instillation and continues for 1 year and 7 months. Disease recurrence with invasive progression occurs 20 months from initial TUR. Four intervening time-points for surveillance cystoscopy/cytology are included without corresponding UroAmplitude testing. Atypical cytology results are noted as reactive (R) or suspicious (S). Low frequency (<1.0%), nonrecurring mutations are withheld in the interest of space and clarity.

## Data Availability

Genome sequencing data used in this study have been deposited in the Sequence Read Archive (SRA) at the National Center for Biotechnology Information (NCBI) and are available through BioProject ID PRJNA817277.

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
