# Peer review of "Urinary Comprehensive Genomic Profiling Correlates Urothelial Carcinoma Mutations with Clinical Risk and Efficacy of Intervention"

_jcm, 2022, doi:10.3390/jcm11195827_

Round 1

Reviewer 1 Report

The authors evaluated how the urinary comprehensive genomic profiling might correlates urothelial carcinoma mutations with clinical risk and efficacy of intervention. The authors stressed the concept of how the clinical standard of care for urothelial carcinoma relies on invasive procedures with suboptimal performance. Hence, the authors developed a urinary comprehensive genomic profiling uCGP test, UroAmplitude, that measures mutations from tumor DNA present in urine, in order to enhance Urothelial Carcinoma treatment. The research is well presented, the paper is well written, the references are correctly reported and are updated, the materials and methods are clearly stated, the results are clearly showed, the discussion is well conducted and conclusion based on data.

Author Response

Dear Reviewer #1, 

We thank you for your careful consideration of our article, “Urinary comprehensive genomic profiling correlates urothelial carcinoma mutations with clinical risk and efficacy of intervention.” Please see our resubmission cover letter for details on the minor revisions made to the manuscript. 

Thank you again 

Trevor Levin

Reviewer 2 Report

This article entitled “Urinary comprehensive genomic profiling correlates urothelial carcinoma mutations with clinical risk and efficacy of intervention ” by Vincent T. Bicocca et al is a very interesting study and provides sufficient information that deserves oncologists' and clinicians’ attention. The authors tried to describe the validation of NGS-based UroAmplitude in urine and compared it to those of matched tumor tissue to validate clinical performance. Overall, the manuscript is well composed. Please describe in more detail the usefulness or benefits and limitations of liquid biopsy in the discussion. Please check the grammar if possible.

Author Response

Dear Reviewer #2, 

We thank you for your careful consideration of our article, “Urinary comprehensive genomic profiling correlates urothelial carcinoma mutations with clinical risk and efficacy of intervention.”   

As requested, we have thoroughly examined the manuscript for improvement of grammar and overall clarity. Regarding concerns with our discussion, we have worked to clarify and elaborate the benefits and limitation of liquid biopsy. We note that our Discussion is not organized with a defined “benefits and limitations” section, and we instead discuss relevant strengths and weaknesses throughout the discussion. However, as you note, key benefits and limitations were missing from the discussion, and we have done our best to redress those oversights.  

A critical weakness that is now addressed in the discussion is the technology’s limited ability to detect rare tumors of the bladder. Because UroAmplitude’s gene panel is designed to identify the most common mutations in UC, non-urothelial cancers of the bladder and rare subtypes of UC are more likely to be missed because they are more likely to be defined by mutations outside the current panel (see line 587). Another key limitation is the remaining reliance on cystoscopy to identify the site of a tumor and the dilemma presented to physicians when a positive UroAmplitude test is identifying a cancer that is not yet visible by cystoscopy (see line 672). Finally, we further emphasize the challenge of genomic studies wherein large-scale measurements also requires large and clinically diverse cohorts (see line 662).  

Thank you again, 

Trevor

Reviewer 3 Report

Dear authors,

I read with interest your article. The methods and statistical analysis is rigorously conducted and your results demonstrate a possible application of the UroAmplitude platform in the diagnosis, treatment and follow-up of the Urothelial Carcinoma.

For the aforementioned reasons, it is my opinion that no further major revisions are needed.

Author Response

Dear Reviewer #3,  

We thank you for your careful consideration of our article, “Urinary comprehensive genomic profiling correlates urothelial carcinoma mutations with clinical risk and efficacy of intervention.” We appreciate your positive reception of our work. Please see our resubmission cover letter for details on the minor revisions made to the manuscript. 

Thank you again,

Trevor